# Clinical effects of dexamethasone among patients with sickle cell disease hospitalized with COVID-19: Outcomes from a single academic health system

**William M. Garneau**[1☯]*, **Matthew J. Lankiewicz**[2☯], **Catherine R. Lesko**[3], **Ashley P. Lauriello**[4], **Kelly A. Gebo**[5], **Sophie M. Lanzkron**[4]

**1** Division of Hospital Medicine, Department of Medicine, Johns Hopkins Hospital, Baltimore, MD, United States of America, **2** Department of Medicine, Johns Hopkins Bayview Hospital, Baltimore, MD, United States of America, **3** Johns Hopkins Bloomberg School of Public Health, Baltimore, MD, United States of America, **4** Department of Medicine, Department of Hematology, Thomas Jefferson University, Philadelphia, PA, United States of America, **5** Division of Infectious Diseases, Department of Medicine, Johns Hopkins Hospital, Baltimore, MD, United States of America

☯ These authors contributed equally to this work.
* william.garneau@jhmi.edu

**Data Availability Statement:** Data cannot be shared publicly because disclosure of the dataset to a third party is not permitted under the approved

## Abstract

### Background

Dexamethasone is a steroid used in the treatment of hospitalized patients with severe COVID-19. However, the effect of dexamethasone in patients with SCD remains unclear given that steroids may precipitate vaso-occlusive crisis (VOC) in patients with SCD.

### Methods and findings

We performed a retrospective analysis of patients with SCD who were hospitalized at Johns Hopkins Health System between June 1, 2020 and June 26, 2022. We reviewed individual charts to assess severity of illness and eligibility for dexamethasone treatment. The exposure of interest was treatment with dexamethasone. Outcomes of interest included incident VTE, length of hospital stay, ICU admission, follow up-VOC and mortality. We identified 30 patients with SCD and COVID-19 who were eligible for dexamethasone treatment, 13 of whom received dexamethasone. Dexamethasone was associated with an increased risk of incident VTE (risk difference = 36%; 95% CI 8%, 66%) after adjustment for high-risk genotypes, >3 hospitalizations, and receipt of anticoagulation. There was an increase in the risk difference of ICU admission and an increased length of stay in crude and adjusted analyses however these associations were not statistically significant.

### Conclusions

We analyzed outcomes among patients with SCD who were hospitalized for COVID-19 and eligible for dexamethasone. Our study suggests that in this population, treatment with dexamethasone increases the risk of incident VTE. There was a suggestion of an increased risk

research protocol. Requests for data access can be submitted to the Johns Hopkins Medicine Institutional Review Board at jhmeirb@jhmi.edu.

**Funding:** This study was funded principally by the NIH National Center for Advancing Translational Sciences (NCATS) KL2TR003099) and the Institute for Clinical and Translational Research (1UL1TR001079-01). The study sponsors did not contribute to the study design, the collection, analysis, and interpretation of data, or the decision to submit this manuscript for publication.

**Competing interests:** I have read the journal's policy and the authors of this manuscript have the following competing interests: KAG receives royalties from UpToDate, non-paid position at Pfizer, and personal consulting from Spark HealthCare, Premier HealthCare, Harrison Consulting and MedEd Learning. WMG reports receiving honorarium from DKBmed; serving as a scientific advisor to Gilead Sciences, Inc; and owning stock in Abbott Laboratories, Danaher, Eli Lilly and Company, Iqvia, Johnson & Johnson, Stryker, UnitedHealth Group, and AstraZeneca Pharmaceuticals. SML reports consulting with bluebird bio, Agios Pharmaceuticals, Novo Nordisk, Pfizer, Merck, GlycoMimetics; prior research support from CSL Behring, Novartis, Takeda Pharmaceuticals; ongoing research support from Patient-Centered Outcomes Research Institute (PCORI), Health Resources and Services Administration and stock ownership in Pfizer, Teva Pharmaceuticals. This does not alter our adherence to PLOS ONE policies on sharing data and materials.

of ICU admission as well as increased length of hospitalization; larger studies are needed to confirm these findings.

## Introduction

Sickle cell disease (SCD) comprises a range of hemoglobinopathies related to a single gene mutation of the hemoglobin molecule [1]. The genetic change alters the structure of the red blood cell leading to reduced oxygen carrying capacity to tissues throughout the body [2]. Patients experience episodic pain related to impaired circulation as well as chronic ischemia and hemolysis resulting in end organ damage, including heart failure, kidney disease, asplenia, and avascular necrosis [2]. Treatment with hydroxyurea to increase fetal hemoglobin expression has long been a mainstay of therapy with new treatments including monoclonal antibody treatment and gene therapy that have recently been approved [2–4].

Patients with SCD may have been especially vulnerable during the COVID-19 pandemic [5]. Although patients with SCD appear to have had a higher risk of hospitalization due to COVID-19, the risk of severe illness, mechanical ventilation or death among hospitalized persons with SCD remains unclear with some evidence of increased risk among certain SCD genotypes and subgroups [5–9]. The routine use of dexamethasone for COVID-19-related hypoxia became standard of care after June 2020 [10]. Since that time, there has been growing recognition that systemic corticosteroids may induce vaso-occlusive crisis in patients with SCD [11, 12].

Due to this concern for steroid treatment causing vaso-occlusive crisis in patients with SCD, there was increased scrutiny of patients with SCD who met criteria for dexamethasone and use in this population varied during the pandemic. The purpose of this study was to evaluate outcomes among patients with SCD hospitalized for SARS-CoV-2 who received dexamethasone.

## Methods

### Study sample

Data for this retrospective cohort study were drawn from the Johns Hopkins CROWN (JH-CROWN) registry [13], an electronic medical record (EMR) that included patients who underwent SARS-CoV-2 testing within the JHHS between June 1, 2020 and June 26, 2022. The research was reviewed by the Johns Hopkins Medicine Institutional Review Board (IRB00328430) and granted a waiver of consent due to the observational nature of the study. The query was conducted on August 23, 2022. Researchers had access to patient-level data to validate patients identified by the query. The JHHS is a network of five hospitals and more than 40 outpatient facilities located in Maryland, Washington, DC, and one pediatric hospital in St. Petersburg, Florida which is not included in the CROWN registry.

Patients were identified using a SQL query for International Classification of Diseases, 10th Revision, Clinical Modification (ICD-10-CM) diagnosis of sickle cell disease ("D57") or any instance of "sickle" in the diagnosis list within the JH-CROWN database. Additional eligibility criteria included 1) age ≥ 18 years; 2) hospitalized with SARS-CoV-2 in the JHHS between June 1, 2020 and June 26, 2022, and 3) qualification for dexamethasone (oxygen saturation ≤ 94%). Patients who received dexamethasone after specified outcomes (admission to the intensive care unit or VTE) were not included. June 1, 2020 was selected as this corresponds to when the RECOVERY trial demonstrated the mortality benefit of dexamethasone

among hospitalized patients with hypoxia due to SARS-CoV-2 infection [14]. Patients with positive SARS-CoV-2 tests outside of JHHS who were subsequently transferred or otherwise treated at JHHS were included if there was documentation of their diagnostic test and treatment at an outside facility For patients with multiple admissions, only the initial admission was included in the analysis.

## Record review

Medical records for patients meeting eligibility criteria were manually reviewed by study team members (WG, ML) and data were abstracted using Research Electronic Data Capture (RED-Cap, version 14.0.10, Vanderbilt University). Differences were resolved via discussion until consensus was reached among reviewers. Age was defined as the age at the time of SARS-CoV-2 diagnosis. Race and sex at birth were abstracted from the demographics recorded in the EMR. Hemoglobin genotype was defined as Sickle-Sickle, Sickle C, Sickle beta plus Thalassemia, and Sickle beta zero Thalassemia. Insurance status was defined as private insurance, Medicare, Medicaid, uninsured or unknown.

Risk of severe illness due to COVID-19 was assessed using the Infectious Disease Society of America (IDSA) list of comorbid conditions [15]. Medical conditions were manually abstracted from chart review. Treatment for vaso-occlusive crisis at time of presentation was defined as receipt of opiates at admission with no other cause of pain identified. Acute chest syndrome at presentation was defined as presence of three factors: 1) fever greater than 100˚C, 2) new infiltrate on chest x-ray or computed tomography of chest, and 3) chest pain. COVID-19 vaccination was considered completed if one-shot of Janssen COVID-19 vaccine or two-shot series with bivalent mRNA dose (Moderna or Pfizer-BioNTech) completed at least two weeks prior to admission. Maintenance therapy for sickle cell disease was recorded from clinical visits prior to hospitalization with COVID-19. Treatment with outpatient COVID-19 therapy was abstracted from the clinical notes. The World Health Organization (WHO) COVID-19 severity score was used to assign severity of illness [16]. The WHO COVID-19 severity score ranges from 0 (uninfected) to 10 (death). Details of inpatient treatment were abstracted from the EMR. Eligibility for dexamethasone was one pulse oximetry reading of oxygen saturation ≤94%.

## Outcomes

The primary outcome of interest was admission to an ICU. Secondary outcomes were hospital length of stay, incidence of venous thromboembolism, follow-up VOC occurring within 30 days of index admission, and in-hospital mortality. Treatment with dexamethasone was defined as at receipt of at least one dose of dexamethasone while hospitalized, prior to VTE or ICU admission. Patients who started dexamethasone prior to any of the clinical outcomes were not included. Critical care was defined by documentation of treatment in an intensive care unit. Venous thromboembolism was defined as any radiologically confirmed acute or chronic venous thromboembolism. Follow-up VOC was defined as an emergency department visit, hospital re-admission, or urgent visit to sickle cell infusion center for VOC within 30 days of index hospitalization for COVID-19.

## Statistical analysis

We stratified the sample by treatment with dexamethasone and compared demographic features including age, sex, race, ethnicity, insurance status, SCD specific comorbid conditions, hemoglobin genotype, maintenance SCD therapy, Infectious Diseases Society of America (IDSA) risk factors, and baseline treatment with anticoagulation.

We described clinical features for patients who did versus did not receive dexamethasone, including WHO severity score on admission, outside hospital transfer status, use of PCA for pain control, outpatient COVID-19 treatments prior to admission, blood transfusions, red cell exchange, highest level of respiratory support and treatment with remdesivir. We used Kruskal Wallis tests for continuous variables and Pearson chi square and Fisher's exact test as appropriate for categorical variables to compare the distribution of demographic and clinical features of patients by receipt of dexamethasone.

Variables reported in the literature that were strongly associated with the risk of individual clinical outcomes such as VTE were controlled in the adjusted analysis [17].

To estimate adjusted risk difference for the association between dexamethasone and ICU admission, length of hospitalization, readmission, and discharge as deceased we adjusted for age, obesity and total number of IDSA risk factors. To estimate adjusted risk difference for the association between dexamethasone and the outcomes of VTE during admission we adjusted for specific VTE risk factors including treatment with AC at time of admission, >3 hospitalizations in preceding year, and high-risk hemoglobin genotype. Adjustment was accomplished by standardization with inverse probability of treatment weights. When estimating the weights, we used a continuity correction to account for the small sample size. We calculated 95% confidence intervals based on the 2.5th and 97.5th percentiles from 400 estimates from nonparametric bootstrap resamples of the data [18]. Data were analyzed using Stata (Stata Statistical Software: Release 18, StataCorp LLC, College Station, TX) and SAS (Statistical Analysis Software Version 9.4, SAS Inc, Cary, NC).

## Results

One hundred twenty-one patients with SCD and a positive SARS-CoV-2 test were identified by the SQL query of the JH-CROWN registry and screened for inclusion. A total of 70 patients with SCD were hospitalized for infection with SARS-CoV-2. Thirty patients were eligible for dexamethasone due to a documented oxygen saturation ≤94% and were included in the analysis (Fig 1). Thirteen patients received dexamethasone and 17 patients did not receive dexamethasone. The median age was 33.7 years old (SD = 11.5), the majority were female (20/30; 67%), black (30/30; 100%), and non-Hispanic or Latino (30/30; 100%). Younger patients were more likely to be treated with dexamethasone than older patients; all other demographic features were similar between the two groups (Table 1).

The median time from admission to dexamethasone initiation was 1 day (range: 0–2 days). The number of IDSA risk factors was similar between dexamethasone vs. non-dexamethasone treated patients (2.4 vs. 2.2; p = 0.06). The most common genotype was Hemoglobin SS (69.2% vs. 70.6%; p = 0.4) followed by Hemoglobin SC (15.4% vs. 17.6%; p = 0.4) among dexamethasone vs. non-dexamethasone treated patients. Allo-immunization was more common in patients who did not receive dexamethasone (15.4% vs. 58.8%; p value 0.03), while history of surgical splenectomy (30.8% vs. 0%; p value 0.03) and presence of an indwelling catheter (46.2% vs. 0%; p value 0.003) were more common in dexamethasone treated group. Other comorbidities related to SCD were similar across dexamethasone vs. non-dexamethasone treated patients. There were no significant differences by maintenance outpatient SCD therapies among the dexamethasone vs. non-dexamethasone treated patients: hydroxyurea (23.1% vs. 35.3%; p = 0.69), simple transfusion (15.4% vs. 5.9%; p = 0.57); red cell exchange (23.1% vs. 5.9%; p = 0.29), voxelotor (0% vs. 11.8%; p value 0.49), crizanlizumab (7.7% vs. 5.9%; p = 1.00), none (30.8% vs. 35.3%; p = 1.00). Additionally, use of anticoagulation prior to admission was similar among dexamethasone vs. non-dexamethasone treated patients (38.5% vs. 29.4%; p = 0.71).

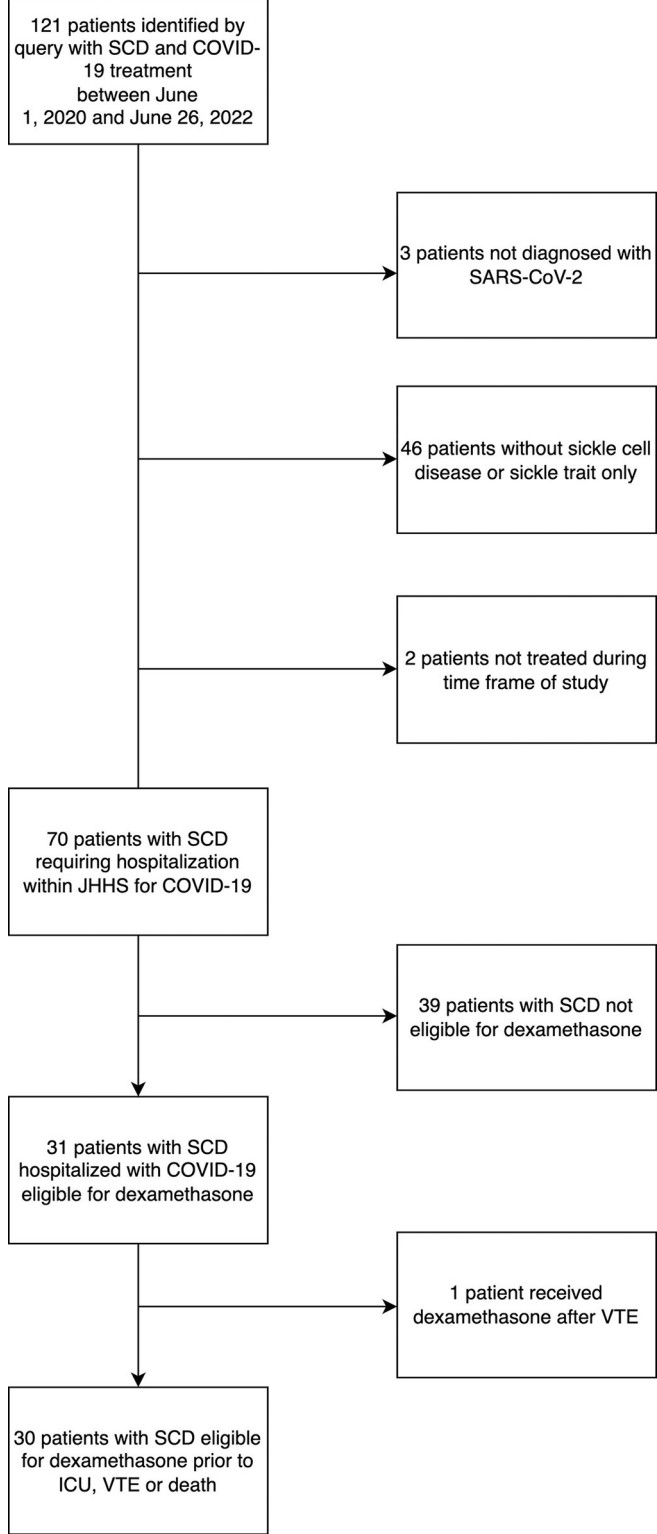

**Fig 1. Cohort flowchart.** Patients with sickle cell disease hospitalized with COVID-19 eligible for dexamethasone treatment within Johns Hopkins Health System June 1, 2020—June 26, 2022.

**Table 1. Demographics and clinical characteristics of patients with sickle cell disease hospitalized for severe COVID-19 with hypoxia.**

| | | Treated with dexamethasone | | |
| --- | --- | --- | --- | --- |
| | | **Yes** | **No** | **P value** |
| **N** | | 13 (43.3%) | 17 (56.7%) | |
| **Age (Mean)** | | 28.6 | 39.9 | 0.002 |
| **Sex** | | | | |
| | Male | 4 (30.8%) | 6 (35.3%) | 1.000 |
| | Female | 9 (69.2%) | 11 (64.7%) | |
| **Race** | | | | |
| | Black or African American | 13 (100.0%) | 17 (100.0%) | - |
| **Ethnicity** | | | | |
| | Not Hispanic or Latino | 13 (100.0%) | 17 (100.0%) | - |
| **Insurance status** | | | | 0.035 |
| | Uninsured | 0 (0.0%) | 1 (5.9%) | |
| | Medicare | 3 (23.1%) | 9 (52.9%) | |
| | Medicaid | 8 (61.5%) | 2 (11.8%) | |
| | Private insurance | 2 (15.4%) | 5 (29.4%) | |
| **Comorbid conditions** | | | | |
| | Pulmonary HTN | 0 (0.0%) | 3 (17.6%) | 0.238 |
| | Secondary hemochromatosis | 6 (46.2%) | 3 (17.6%) | 0.123 |
| | Venous thromboembolism | 6 (46.2%) | 7 (41.2%) | 1.000 |
| | Cerebrovascular accident | 2 (15.4%) | 1 (5.9%) | 0.565 |
| | Acute chest syndrome | 9 (69.2%) | 12 (70.6%) | 1.000 |
| | Avascular necrosis | 2 (15.4%) | 7 (41.2%) | 0.229 |
| | Allo-immunization | 2 (15.4%) | 10 (58.8%) | 0.026 |
| | Priapism | 1 (7.7%) | 1 (5.9%) | 1.000 |
| **Outpatient anticoagulation** | | | | |
| | Yes | 5 (38.5%) | 5 (29.4%) | 0.705 |
| **>3 hospitalizations/year** | | | | |
| | Yes | 4 (30.8%) | 2 (11.8%) | 0.360 |
| **Indwelling catheter** | | | | |
| | Yes | 6 (46.2%) | 0 (0.0%) | 0.003 |
| **Splenectomy** | | | | |
| | Yes | 4 (30.8%) | 0 (0.0%) | 0.026 |
| **Hemoglobin genotype** | | | | |
| | SS | 9 (69.2%) | 12 (70.6%) | 0.373 |
| | S-Beta-Thal0 | 0 (0.0%) | 2 (11.8%) | |
| | S-Beta-Thal+ | 2 (15.4%) | 0 (0.0%) | |
| | Hemoglobin SC | 2 (15.4%) | 3 (17.6%) | |
| **Maintenance SCD therapy** | | | | |
| | Hydroxyurea | 3 (23.1%) | 6 (35.3%) | 0.691 |
| | Simple transfusion | 2 (15.4%) | 1 (5.9%) | 0.565 |
| | RCE | 3 (23.1%) | 1 (5.9%) | 0.290 |
| | Crizanlizumab | 1 (7.7%) | 1 (5.9%) | 1.000 |
| | Voxelotor | 0 (0.0%) | 2 (11.8%) | 0.492 |
| | None | 4 (30.8%) | 6 (35.3%) | 1.000 |
| | Unknown | 1 (7.7%) | 0 (0.0%) | 0.433 |
| **Vaccination status[a]** | | | | |
| | Unvaccinated | 11 (84.6%) | 11 (64.7%) | 0.564 |

(*Continued*)

**Table 1.** (Continued)

|  |  | Treated with dexamethasone | | |
| --- | --- | --- | --- | --- |
|  |  | **Yes** | **No** | **P value** |
|  | Vaccinated | 2 (15.4%) | 4 (23.5%) |  |
|  | Vaccinated + booster | 0 (0.0%) | 2 (11.8%) |  |
| **IDSA risk factors**[b] |  |  |  |  |
|  | 1 | 2 (15.4%) | 7 (41.2%) | 0.066 |
|  | 2 | 8 (61.5%) | 3 (17.6%) |  |
|  | 3 | 1 (7.7%) | 4 (23.5%) |  |
|  | 4 | 1 (7.7%) | 3 (17.6%) |  |
|  | 5 | 0 (0.0%) | 0 (0.0%) |  |
|  | 6 | 0 (0.0%) | 0 (0.0%) |  |
|  | 7 | 1 (7.7%) | 0 (0.0%) |  |

[a] Vaccination series completed >2 weeks prior to admission

[b] Age ≥65 years, BMI >25 kg/m2 Pregnancy, chronic kidney disease, diabetes mellitus, immunosuprressing medication, cardiovascular disease or hypertension, chronic lung disease, sickle cell disease, neurodevelopmental disorder, medical technological dependence

Rate of vaccination were similar among dexamethasone vs. non-dexamethasone treated patients. Only one patient had received outpatient therapy with a monoclonal antibody prior to admission, and no patients received molnupiravir, remdesivir, or nirmatrelvir/ritonavir prior to admission (Table 2).

WHO severity score on admission was similar between dexamethasone vs. non-dexamethasone treated patients (5.3 vs. 4.8; p = 0.23). There were no differences in patients who were outside hospital transfers in terms of those who were treated with dexamethasone vs. non-dexamethasone (15.4% vs. 0% p = 0.18). Treatment with simple transfusion among dexamethasone vs. non-dexamethasone treated patients (61.5% vs. 35.3%; p = 0.27) was similar. A higher percentage of dexamethasone vs. non-dexamethasone treated patients received red cell exchange (38.5% vs. 5.9%; p = 0.06) but the difference was not statistically significant. Remdesivir use was similar between dexamethasone vs. non-dexamethasone treated patients (92.3% vs. 70.6%; p = 0.20).

There was higher incidence of VTE during admission among dexamethasone vs. non-dexamethasone treated patients (30.8% vs. 0%; p value = 0.03). Length of stay was longer in patients who received dexamethasone but did not reach statistical significance (14.0 vs 5.5 days, p value = 0.06). More patients who received dexamethasone vs. non-dexamethasone were admitted to the ICU (30.8% vs. 5.9%; p value 0.14). There was a similar rate of VOC in patients who received dexamethasone compared to those who did not (7.7% vs. 17.6%; p value 0.613). One patient in each group died during their hospitalization (7.7% vs. 5.9%; p value = 1.000) (Table 3).

The risk of admission to ICU was higher for patients treated with dexamethasone compared to those who were not treated: risk difference = 25% (95% CI -3%, 47%). The risk difference was 17% (95% CI -36%, 53%) after adjusting for age, obesity and total number of IDSA risk factors. Risk of VTE was higher for patients treated with dexamethasone during admission: crude risk difference = 31% (95% CI 7%, 53%) and risk difference = 36% (95% CI 8%, 66%) after adjustment for high-risk sickle cell genotype, >3 hospitalizations per year and receipt of anticoagulation. The risk of readmission to the hospital was lower for patients treated with dexamethasone during admission: crude risk difference -10% (95% CI -33%, 13%) and risk difference = -5% (95% CI -59%, 39%) after adjusting for age, obesity and total number of IDSA risk factors.

**Table 2. Clinical care of patients with sickle cell disease hospitalized for severe COVID-19 with hypoxia.**

| | | Treated with dexamethasone | | |
|---|---|---|---|---|
| | | Yes | No | P value |
| N | | 13 (43.3%) | 17 (56.7%) | |
| **WHO severity score on admission** | | 5.3 | 4.8 | 0.233 |
| **Day of hospitalization received dexamethasone** | | | | - |
| | 1 | 6 (46.2%) | 0 (0.0%) | |
| | 2 | 6 (46.2%) | 0 (0.0%) | |
| | 3 | 1 (7.7%) | 0 (0.0%) | |
| **Outside hospital transfer** | | | | |
| | Yes | 2 (15.4%) | 0 (0.0%) | 0.179 |
| **PCA for pain control** | | | | |
| | Yes | 8 (61.5%) | 11 (64.7%) | 1.000 |
| **Outpatient COVID-19 treatments** | | | | |
| | Monoclonal antibodies | 0 (0.0%) | 1 (5.9%) | 1.000 |
| | None | 13 (100.0%) | 16 (94.1%) | |
| | **Received simple PRBC transfusion** | 8 (61.5%) | 6 (35.3%) | 0.269 |
| **Number of units** | | | | |
| | 1 | 1 (12.5%) | 1 (16.7%) | 0.720 |
| | 2 | 4 (50.0%) | 5 (83.3%) | |
| | 3 | 0 (0.0%) | 0 (0.0%) | |
| | 4 | 2 (25.0%) | 0 (0.0%) | |
| | 5 | 0 (0.0%) | 0 (0.0%) | |
| | 6 | 1 (12.5%) | 0 (0.0%) | |
| | **Red cell exchange** | 5 (38.5%) | 1 (5.9%) | 0.061 |
| **Any transfusion (red cell exchange or simple)** | | | | |
| | Yes | 10 (76.9%) | 7 (41.2%) | 0.071 |
| **Treated with remdesivir** | | | | |
| | Yes | 12 (92.3%) | 12 (70.6%) | 0.196 |
| **Highest level of respiratory support** | | | | |
| | None | 0 (0.0%) | 3 (17.6%) | 0.036 |
| | Nasal canula | 10 (76.9%) | 14 (82.4%) | |
| | Mechanical ventilation | 3 (23.1%) | 0 (0.0%) | |

WHO: World Health Organization, PCA: patient-controlled analgesia, PRBC: packed red blood cells

**Table 3. Clinical outcomes for patients with sickle cell disease hospitalized with severe COVID-19.**

| | Treated with dexamethasone | Treated with dexamethasone | Treated with dexamethasone |
|---|---|---|---|
| | Yes | No | P value |
| N | 13 (43.3%) | 17 (56.7%) | |
| **Mean length of hospitalization (days)** | 14.0 | 5.5 | 0.063 |
| **VTE during admission** | 4 (30.8%) | 0 (0.0%) | 0.026 |
| **ICU admission** | 4 (30.8%) | 1 (5.9%) | 0.138 |
| **VOC within 30 days of admission[1]** | 1 (7.7%) | 3 (17.6%) | 0.613 |
| **Discharged as deceased** | 1 (7.7%) | 1 (5.9%) | 1.000 |

ICU = intensive care unit, VTE = venous thromboembolism, VOC = vaso-occlusive crisis

[1] Emergency department visit, hospital re-admission, urgent visit to sickle cell infusion center for VOC within 30 days of index hospitalization

**Table 4. Adjusted and unadjusted risk difference for clinical outcomes and difference in length of stay among patients with sickle cell disease hospitalized with severe COVID-19.**

|  | Treated with dexamethasone | |  |  |
|---|---|---|---|---|
|  | Yes | No | Difference, crude | Difference, adjusted[a] |
| ICU admission[b] | 31% (7%, 51%) | 6% (0%, 19%) | 25% (-3%, 47%) | 17% (-36%, 53%) |
| Length of hospitalization[b] | 14.0 (6.0, 21.3) | 5.5 (3.7, 8.5) | 8.5 (-0.7, 16.7) | 4.4 (-3.1, 13.6) |
| VTE during admission[c] | 31% (8%, 54%) | 0% (0%, 2%) | 31% (7%, 53%) | 36% (8%, 66%) |
| Readmission[b] | 8% (0, 25%) | 18% (1%, 35%) | -10% (-33%, 13%) | -5% (-59%, 39%) |
| Discharged as deceased[b] | 8% (0, 21%) | 6% (0%, 19%) | 2% (-17%, 19%) | 12% (-10%, 41%) |

[a] Adjustment was carried out with standardization; 95% confidence intervals are the 2.5th and 97.5th percentiles from 400 estimates from non-parametric bootstrap resamples of the data with a continuity correction applied to account for small cell counts.

[b] Adjusted for age, obesity, and total number of IDSA risk factors

[c] Adjusted for high-risk genotypes, >3 hospitalizations, and receipt of anticoagulation

For the model evaluating risk of VTE with dexamethasone, inclusion of indwelling catheter or splenectomy caused the model to fail to converge; these variables were excluded from the multivariable analysis. The risk of death was similar for patients treated with dexamethasone versus not: crude risk difference = 2% (95% CI -17%, 19%) and adjusted risk difference = 12% (95% CI -10%, 41%), adjusted for age, obesity and total number of IDSA risk factors (Table 4).

## Discussion

This study has several important findings. First, dexamethasone treatment was associated with subsequent VTE in patients with SCD and hypoxia due to SARS-CoV-2. Second, our findings suggest a trend toward increased risk of admission to the ICU and length of stay amongst the patients who received steroids. Lastly, there was no significant association of dexamethasone treatment and mortality.

Our findings are consistent with moderate harm of dexamethasone for patients with SCD and hypoxia due to SARS-CoV-2. Due to the observational nature of the study, there may be uncontrolled confounding by indication. While allo-immunization, indwelling catheter and splenectomy were more common in the dexamethasone-treatment group and may contribute to the higher rates of VTE; the dexamethasone-treatment group were younger (28.6 vs. 39.9 years old) and had a higher overall treatment with simple transfusion (61.5% vs. 35.3%) as well as red cell exchange (38.5% and 5.9%); all of which would tend to skew the results towards improved outcomes in this group. Overall, the absence of a clear benefit and suggestion of harm of dexamethasone for patients with SCD indicate further investigation is needed.

These findings should be considered within the limited body of literature examining dexamethasone use for COVID-19 and sickle cell disease. One of the largest cohort studies of 609 hospitalized patients with SCD treated for COVID-19 reported that 24% received dexamethasone but associations between treatment and subsequent clinical outcome was not assessed [19]. In a small case series of 31 pediatric patients with SCD, the authors reported that three children treated with dexamethasone did not develop severe illness [20].

This study found an increased risk of VTE in patients with SCD who were eligible for dexamethasone and received corticosteroids. This should be interpreted with caution as we were unable to include indwelling catheter or splenectomy as covariates in the risk difference which may account for the increased risk of VTE seen in the dexamethasone-treated population. Additionally, there were higher rates of transfusion in the dexamethasone-treated cohort; the group was also younger and had more severe illness at admission. Thus, while we have

attempted to isolate the effect of dexamethasone there is likely significant persistent residual confounding. Nonetheless, the finding of increased risk of VTE is particularly notable given relatively low rates of VTE have been reported for patients with SCD hospitalized with COVID-19 [21–23]. These findings suggest there may be a relationship between the action of steroids in patients with SCD that predisposes patients to VTE; however this should be studied in larger cohorts.

We observed a non-significant association of mortality with treatment with dexamethasone. The lack of statistical significance is likely related to the low event rate overall: only two patients among the 30 who were eligible for dexamethasone died during hospitalization. These findings are in keeping with a growing recognition of the risk of steroids in this population. A nationwide cohort study conducted in France recently reported increased risk of hospitalization for vaso-occlusive crisis associated with steroid exposure [11]. The present study is the first to specifically examine patients with SCD who were eligible for dexamethasone and associated clinical outcomes due to COVID-19. There is not yet a consensus regarding the role of dexamethasone in COVID-19 among patients with SCD–while providers at Johns Hopkins Hospital did not routinely administer steroids, this is not reflected in guidance from the American Society of Hematology [24]. Larger observational studies are needed to address this question fully and to further assess causality.

This study has strengths, including the use of WHO severity score, SCD comorbid conditions, hemoglobin type, IDSA risk factors for severe COVID-19, pre-hospital, and hospital treatments for COVID-19, SCD therapy prior to admission, outside hospital transfer status in addition to other variables to assess comparability. However, this study also has limitations. First, the small sample size did not allow for extensive adjustment of potential confounders. There was also a higher percentage of patients who were transferred from outside hospitals who received dexamethasone and also a higher percentage of patients who underwent red cell exchange who were treated with dexamethasone. These differences may confound associations between dexamethasone and the outcomes of interest. Additionally, this study reported outcomes from an academic medical center in the United States with an inpatient treatment service for sickle cell patients; these results may not generalize to other clinical systems and countries.

Results from this study must be interpreted with caution. Larger studies would strengthen the research on risk of dexamethasone in this population. Additionally, these findings were among a primarily unvaccinated cohort of patients and the effects of dexamethasone in patients with prior history of covid infection or vaccination may be different and merit exploration. As COVID-19 continues to infect patients with SCD, clinicians should be mindful of the risks associated with treatment; other researchers should investigate this association with larger studies to explore this effect. This is an important area for future research given the vulnerability of patients with SCD and the ongoing threat of SARS-CoV-2 infection.

## Author Contributions

**Conceptualization:** William M. Garneau, Matthew J. Lankiewicz, Ashley P. Lauriello, Kelly A. Gebo, Sophie M. Lanzkron.

**Data curation:** William M. Garneau, Matthew J. Lankiewicz, Kelly A. Gebo, Sophie M. Lanzkron.

**Formal analysis:** William M. Garneau, Catherine R. Lesko, Kelly A. Gebo, Sophie M. Lanzkron.

**Funding acquisition:** William M. Garneau.

**Investigation:** William M. Garneau, Matthew J. Lankiewicz, Catherine R. Lesko, Ashley P. Lauriello, Kelly A. Gebo, Sophie M. Lanzkron.

**Methodology:** William M. Garneau, Matthew J. Lankiewicz, Catherine R. Lesko, Kelly A. Gebo, Sophie M. Lanzkron.

**Project administration:** William M. Garneau, Sophie M. Lanzkron.

**Resources:** William M. Garneau, Kelly A. Gebo.

**Software:** William M. Garneau, Catherine R. Lesko.

**Supervision:** William M. Garneau, Kelly A. Gebo, Sophie M. Lanzkron.

**Validation:** William M. Garneau, Matthew J. Lankiewicz, Catherine R. Lesko, Kelly A. Gebo.

**Visualization:** William M. Garneau.

**Writing – original draft:** William M. Garneau, Matthew J. Lankiewicz, Kelly A. Gebo, Sophie M. Lanzkron.

**Writing – review & editing:** William M. Garneau, Matthew J. Lankiewicz, Catherine R. Lesko, Ashley P. Lauriello, Kelly A. Gebo, Sophie M. Lanzkron.

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
