## [Decision Letter · Decision Letter 0]

9 Sep 2024

PONE-D-24-33591Clinical effects of dexamethasone among patients with sickle cell disease hospitalized with COVID-19: Outcomes from a single academic health systemPLOS ONE

Dear Dr. Garneau,

Thank you for submitting your manuscript to PLOS ONE. After careful consideration, we feel that it has merit but does not fully meet PLOS ONE’s publication criteria as it currently stands. Therefore, we invite you to submit a revised version of the manuscript that addresses the points raised during the review process. Areas to particularly focus on in the revision include clarifying the rates of vaso-occlusive episodes after discharge in those treated with versus without dexamethasone and updating the limitations to include possible confounders for venous thromboembolism in this cohort.

We look forward to receiving your revised manuscript.

Kind regards,

Santosh L. Saraf

Academic Editor

PLOS ONE

Journal Requirements: 

"I have read the journal's policy and the authors of this manuscript have the following competing interests: KAG receives royalties from UpToDate, non-paid position at Pfizer, and personal consulting from Spark HealthCare, Premier HealthCare, Harrison Consulting and MedEd Learning. WMG reports receiving honorarium from DKBmed; serving as a scientific advisor to Gilead Sciences, Inc; and owning stock in Abbott Laboratories, Danaher, Eli Lilly and Company, Iqvia, Johnson & Johnson, Stryker, UnitedHealth Group, and AstraZeneca Pharmaceuticals. SML reports consulting with bluebird bio, Agios Pharmaceuticals, Novo Nordisk, Pfizer, Merck, GlycoMimetics; prior research support from CSL Behring, Novartis, Takeda Pharmaceuticals; ongoing research support from Patient-Centered Outcomes Research Institute (PCORI), Health Resources and Services Administration and stock ownership in Pfizer, Teva Pharmaceuticals."

5. We note you have included a table to which you do not refer in the text of your manuscript. Please ensure that you refer to Table 3 and 4 in your text; if accepted, production will need this reference to link the reader to the Table.

Reviewers' comments:

Reviewer's Responses to Questions

**Comments to the Author**

1. Is the manuscript technically sound, and do the data support the conclusions?

Reviewer #1: Partly

Reviewer #2: Yes

2. Has the statistical analysis been performed appropriately and rigorously? 

Reviewer #1: Yes

Reviewer #2: Yes

3. Have the authors made all data underlying the findings in their manuscript fully available?

Reviewer #1: Yes

Reviewer #2: Yes

4. Is the manuscript presented in an intelligible fashion and written in standard English?

Reviewer #1: Yes

Reviewer #2: Yes

5. Review Comments to the Author

Reviewer #1: The authors present a thorough retrospective cohort study describing the clinical outcomes of patients with sickle cell disease admitted with COVID, comparing those treated or not treated with dexamethasone.

As they state in the introduction, there is a question among hematologists about the risk of rebound vaso-occlusive episodes of pain after steroid treatment in this population. I do not see results that report on the rates of rebound VOE in this cohort. The authors do state that the rate of readmission among those treated with dex is lower, but it appears that this is all-cause readmission. Please add the rates of VOE after dexamethasone between the groups

The high rate of VTE is an interesting finding, however, it is at high-risk of confounding by indication, as the authors briefly mention in the discussion. More severe COVID, ICU stay, and prolonged hospitalization are all risk factors for VTE, and more severe COVID could also be the reason patients were treated with dexamethasone. It is a strength that the authors include severity score on admission, but patients with COVID can devolve dramatically during hospitalization. The fact that the dexamethasone-treated patients also were more likely to need a transfusion also supports that they were more severe. The authors should elaborate further on this important consideration/limitation in the discussion

Reviewer #2: Thank you for allowing me to review the paper enClinical effects of dexamethasone among patients with sickle cell disease hospitalized with COVIDS-19. Outcomes from a single academic health system." The paper is well-written.

In the results section. the authors note that "Younger patients were more likely to be treated with dexamethasone than older patients. Can the authors proffer a reason for this observation?

6. PLOS authors have the option to publish the peer review history of their article (what does this mean?). If published, this will include your full peer review and any attached files.

Reviewer #1: No

Reviewer #2: No

---

## [Author Response · Author response to Decision Letter 0]

8 Oct 2024

Dear Editorial Team,

With my co-authors, I am pleased to re-submit with revisions our manuscript titled Clinical effects of dexamethasone among patients with sickle cell disease hospitalized with COVID-19: Outcomes from a single academic health system with our response to reviewers’ comments. 

Below are our point-by-point responses. We have bolded the editorial requirements and reviewers’ comments and placed our responses underneath.

Journal Requirements:

The manuscript has been edited to reflect the style requirements

"I have read the journal's policy and the authors of this manuscript have the following competing interests: KAG receives royalties from UpToDate, non-paid position at Pfizer, and personal consulting from Spark HealthCare, Premier HealthCare, Harrison Consulting and MedEd Learning. WMG reports receiving honorarium from DKBmed; serving as a scientific advisor to Gilead Sciences, Inc; and owning stock in Abbott Laboratories, Danaher, Eli Lilly and Company, Iqvia, Johnson & Johnson, Stryker, UnitedHealth Group, and AstraZeneca Pharmaceuticals. SML reports consulting with bluebird bio, Agios Pharmaceuticals, Novo Nordisk, Pfizer, Merck, GlycoMimetics; prior research support from CSL Behring, Novartis, Takeda Pharmaceuticals; ongoing research support from Patient-Centered Outcomes Research Institute (PCORI), Health Resources and Services Administration and stock ownership in Pfizer, Teva Pharmaceuticals."

The statement has been added to a revised cover letter.

The data set for this study includes protected health information and was obtained under a waiver of consent. The study was authorized by the Johns Hopkins Medicine Institutional Review Board and use of the data set is controlled by legal restrictions. Disclosure of the de-identified dataset to a third party is not permitted under the approved research protocol. Requests for data access can be submitted to the Institutional Review Board at jhmeirb@jhmi.edu.

b) If there are no restrictions, please upload the minimal anonymized data set necessary to replicate your study findings to a stable, public repository and provide us with the relevant URLs, DOIs, or accession numbers. For a list of recommended repositories, please see https://journals.plos.org/plosone/s/recommended-repositories. You also have the option of uploading the data as Supporting Information files, but we would recommend depositing data directly to a data repository if possible.

The manuscript has been edited to include the ethics statement in the methods section only.

5. We note you have included a table to which you do not refer in the text of your manuscript. Please ensure that you refer to Table 3 and 4 in your text; if accepted, production will need this reference to link the reader to the Table.

Reference to Table 3 and Table 4 has been added to the manuscript.

Reviewer Comments:

Reviewer #1: The authors present a thorough retrospective cohort study describing the clinical outcomes of patients with sickle cell disease admitted with COVID, comparing those treated or not treated with dexamethasone.

As they state in the introduction, there is a question among hematologists about the risk of rebound vaso-occlusive episodes of pain after steroid treatment in this population. I do not see results that report on the rates of rebound VOE in this cohort. The authors do state that the rate of readmission among those treated with dex is lower, but it appears that this is all-cause readmission. Please add the rates of VOE after dexamethasone between the groups

Thank you for highlighting this omission. A sentence summarizing these findings has been added to the manuscript (Results, Paragraph 5). This information is also listed in Table 3. 

The high rate of VTE is an interesting finding, however, it is at high-risk of confounding by indication, as the authors briefly mention in the discussion. More severe COVID, ICU stay, and prolonged hospitalization are all risk factors for VTE, and more severe COVID could also be the reason patients were treated with dexamethasone. It is a strength that the authors include severity score on admission, but patients with COVID can devolve dramatically during hospitalization. The fact that the dexamethasone-treated patients also were more likely to need a transfusion also supports that they were more severe. The authors should elaborate further on this important consideration/limitation in the discussion

The following statement has been added to the discussion to reflect this point: “Additionally, while we have attempted to control for severity of illness, there may be residual confounding by indication, as evidenced by higher rates of transfusion in this cohort that contribute to the higher rate of VTE in the dexamethasone treated group” (Discussion, Paragraph 4).

Reviewer #2: Thank you for allowing me to review the paper Clinical effects of dexamethasone among patients with sickle cell disease hospitalized with COVIDS-19. Outcomes from a single academic health system." The paper is well-written.

In the results section. the authors note that "Younger patients were more likely to be treated with dexamethasone than older patients. Can the authors proffer a reason for this observation?

Thank you for drawing attention to this finding. In our study, younger patients had more severe COVID-19 as measured by respiratory requirements at admission and overall WHO severity score. Commentary has been added to the discussion (Discussion, Paragraph 4).

End of Comments.

We appreciate the comments of the editorial team and reviewers which have strengthened the manuscript. We look forward to your review of our response. Please do not hesitate to contact me with any questions.

---

## [Editor Report · Decision Letter 1]

22 Oct 2024

Clinical effects of dexamethasone among patients with sickle cell disease hospitalized with COVID-19: Outcomes from a single academic health system

PONE-D-24-33591R1

Dear Dr. Garneau,

We’re pleased to inform you that your manuscript has been judged scientifically suitable for publication and will be formally accepted for publication once it meets all outstanding technical requirements.

Kind regards,

Santosh L. Saraf

Academic Editor

PLOS ONE

---

## [Editor Report · Acceptance letter]

15 Nov 2024

PONE-D-24-33591R1 

PLOS ONE

Dear Dr. Garneau, 

I'm pleased to inform you that your manuscript has been deemed suitable for publication in PLOS ONE. Congratulations! Your manuscript is now being handed over to our production team.

Kind regards, 

on behalf of

Dr. Santosh L. Saraf 

Academic Editor

PLOS ONE